# Congenital CMV, Lights and Shadows on Its Management: The Experience of a Reference Center in Northern Italy

**DOI:** 10.3390/children9050655

**Published:** 2022-05-03

**Authors:** Valeria Rubinacci, Mara Fumagalli, Giulia Meraviglia, Laura Gianolio, Anna Sala, Marta Stracuzzi, Anna Dighera, Gian Vincenzo Zuccotti, Vania Giacomet

**Affiliations:** 1Department of Pediatric Infectious Diseases, Luigi Sacco Hospital, University of Milan, 20122 Milan, Italy; valeria.rubinacci@unimi.it (V.R.); mara.fumagalli@unimi.it (M.F.); giulia.meraviglia@unimi.it (G.M.); laura.gianolio@unimi.it (L.G.); anna.sala1@unimi.it (A.S.); marta.stracuzzi@unimi.it (M.S.); anna.dighera@unimi.it (A.D.); gianvincenzo.zuccotti@unimi.it (G.V.Z.); 2Department of Pediatrics, Vittore Buzzi Hospital, University of Milan, 20122 Milan, Italy

**Keywords:** cytomegalovirus (CMV)_1_, congenital CMV_2_, valganciclovir_3_, TORCH_4_, sensorineural hearing loss (SNHL)_5_

## Abstract

Congenital cytomegalovirus infection (cCMV) is the most common congenital viral infection, with a consistent rate of morbidity, mortality, and long-term sequelae, especially in the case of late diagnosis. Nevertheless, a universal screening for CMV is not currently recommended, and global awareness about this infection, as well as accurate and shared indications on follow-up and treatment, are still lacking. We reviewed data about 59 suspect cCMV cases who referred to our center from 2014 to 2021. We report 41 cases of confirmed cCMV diagnosed at birth, with clinical or radiological abnormalities in 36.6% of them. Other five patients received a late diagnosis and all presented neurological impairment. Twelve patients received therapy with Valganciclovir within the first month of life, with favorable outcome in nine cases. Therapy after the first month of life was attempted in four patients, with improvement in one case. The overall awareness about cCMV infection was 32.6%. Considering our population, maternal serological screening followed by targeted testing of neonates could be an effective strategy. Some aspects of cCMV infection management should be further investigated, such as indication of treatment after the first month of life or in asymptomatic patients. Awareness about the infection should be improved to implement preventive strategies.

## 1. Introduction

Congenital cytomegalovirus infection (cCMV) is the most common congenital viral infection. Its average estimated incidence at birth in high-income countries is 0.64% [1,2]. The incidence rates in Italy are between 0.57% and 1% of all live births [3,4]. Despite its significant prevalence, morbidity, mortality and long-term sequelae, inadequate awareness is still rampant, not only in the general population but also among healthcare workers and pregnant women, as demonstrated by various surveys [5,6,7].

Several studies proved that cCMV prevalence at birth increases with maternal seroprevalence [1], therefore depending on CMV seroprevalence in the general population. In Italy, the seroprevalence of CMV may be as high as 70–80% [3], which indicates that the risk for congenital CMV infection may be of concern. Nevertheless, the universal cytomegalovirus serological screening of pregnant women is not recommended by national public health bodies in any country [8,9].

Literature data indicate that only 10–15% of cCMV newborns are symptomatic at birth [9,10,11,12,13]. Although long-term sequelae are more common among symptomatic patients, concerning around 50% of the cases, according to different studies, they occur also in 8–15% of asymptomatic subjects, especially hearing loss and neurocognitive abnormalities [9,10,11,12,14,15]. Indeed, cCMV represents the leading non-genetic cause of pediatric sensorineural hearing loss (SNHL) and neurocognitive impairment [14]. For this reason, in the literature there is a high level of agreement on the necessity of a long-term follow-up for all infected children, regardless of the symptoms or treatment [9,14].

Even if no antiviral drug is currently licensed for the treatment of cCMV, Valganciclovir is considered a first-line therapy to be administered within the first month of life to moderately to severely symptomatic cases [8,9]. Various case reports and case series, but only two randomized control trials (RCTs), have reported on the treatment of cCMV infection, attesting its efficacy in improving long-term audiologic and neurodevelopmental outcomes in cCMV-infected infants [13,16,17]. However, univocal guidelines on indications and duration of this antiviral therapy for cCMV infection are not yet available [9].

Here, we performed a retrospective review of a cohort of children with confirmed CMV infection afferent to a reference center in Northern Italy, over a period of 7 years. Our aims were to describe the presentation of cCMV infection at birth and to underline the benefits of an antiviral treatment as well as the consequences of a late diagnosis. Secondly, we evaluated the awareness of this disease before pregnancy, underlying its role in reducing the burden of infection.

## 2. Materials and Methods

We retrospectively evaluated the medical records of 59 children referred to the Pediatric Infectious Disease Department of Luigi Sacco Hospital (Milan, Italy) for maternal CMV seropositivity during pregnancy or for clinical suspicion of cCMV, from January 2014 to May 2021. cCMV was confirmed in 46 of 59 children, defined by two positive CMV-DNA quantitative Polymerase Chain Reaction (PCR) detections on urine in the first days of life or one positive CMV-DNA quantitative PCR detection on stored DBS (dried-blood spot). The remaining 13 children tested negative and thus did not undergo other diagnostic procedures, being excluded from further statistical analysis.

All prenatal, perinatal and postnatal histories were reviewed, extracting data about pregnancy, birth, awareness regarding the infection before pregnancy, cCMV diagnosis and related symptoms, diagnostic procedures, eventual antiviral therapy, adverse effects and long-term sequelae follow-up.

The management of these patients was in accordance with the hospital protocol. Complete physical examination, anthropometric measurements, blood tests, brain and abdominal ultrasound examination, neurological examination, audiological screening, fundus oculi examination were performed in the first days of life in all infected children, except those referred to our center after a late diagnosis. Brain Magnetic Resonance Imaging (MRI) was performed in patients with a pathological brain ultrasound, with clinically detectable neurologic findings or with a late diagnosis. Treatment with Valganciclovir (16 mg/kg per dose orally twice a day, for 6 months) was administered in selected cases. For all patients, regardless of their symptoms or treatment, a strict neurological (with psychomotor development evaluation), audiological, ophthalmological and hematochemical follow-up was scheduled up to 6 years of age.

For the purpose of this study, patients were considered asymptomatic or symptomatic with “mild”, “moderate” or “severe” disease according to the latest European Expert Consensus Statement on Diagnosis and Management of Congenital Cytomegalovirus [9].

Descriptive analyses were performed using SPSS statistics ver. 26.0 (IBM, Chicago, IL, USA). All data are presented as medians and percentages.

The study was conducted in accordance with the Declaration of Helsinki and approved by the Ethics Committee of the coordinating center in Milan (protocol number 2020/ST/061 of 21/04/2020).

## 3. Results

We included in the study 46 children with confirmed cCMV infection (M/F = 22:24); 42 of them (91.3%) were newborns at term, the remaining 4 were late-preterm.

In 41 of the 46 patients, diagnosis was made at birth through detection of CMV DNA on urine; testing for CMV at birth was prompted in 40 of the 41 newborns by maternal serology. Indeed, seroconversion during pregnancy was found in 34 cases (82.9% of the total: 41.2% in the first trimester, 41.2% in the second trimester, 17.6% in the third trimester), while in 6 cases (14.6%), serology was consistent with a non-primary maternal infection (reactivation or infection by a different strain); of them, only 2 presented symptoms at birth. In one case, maternal serology was never tested during pregnancy but since it was a neglected pregnancy and congenital toxoplasmosis was also found, the neonate was tested for other congenital infections, and CMV DNA in urine resulted positive.

Late referral to our center occurred for the remaining five patients, leading to a diagnosis at a median age of 9 months (range 3 months–4 years) through the detection of CMV-DNA on stored DBS. Four of these infants were born from mothers with an undetected non-primary infection, while in the remaining case, CMV serology was never tested during pregnancy; neurological signs and hearing impairment were the first clues which led to the diagnosis.

Among the 41 children diagnosed at birth, only 7 cases (17%) had an abnormal clinical examination. However, adding the results of radiological and laboratory investigations, a total of 15 patients (36.6%) were considered symptomatic, showing various degrees of the disease: 2 cases with mild disease, 2 with moderate disease, and 11 with severe disease. The symptoms and signs present at birth are reported in Table 1. Nevertheless, three patients showed cardiac anomalies, which are not a defined manifestation of cCMV: two patients had isolated interventricular septum defects, while one had cardiomegaly and right ventricular hypertrophy.

All the five patients who had a late diagnosis presented neurological impairment (including hemiparesis, microcephaly, epilepsy or psychomotor delay) and pathological MRI findings; in two of them, severe unilateral SNHL was also present.

No cases of CMV retinopathy were reported, neither at birth nor during follow-up.

Considering antiviral therapy, Valganciclovir was administered within the first month of life to 12 of the 13 moderately to severely symptomatic newborns (1 patient with severe disease at birth was lost at follow-up after the first visit and so excluded from further analysis). The reasons for starting this therapy were the following: central nervous system (CNS) involvement in seven cases (58.3%), isolated SNHL in three cases (25%) and hematological alterations in the remaining two (16.7%).

In four patients, treatment with Valganciclovir was started after the first month of life at a median age of 8 months (range 3 months–4 years). Three of them received a late diagnosis and presented with neurological impairment and abnormal neuroimaging (plus SNHL in two cases). In these patients, therapy was discontinued after a median period of 16 weeks due to lack of efficacy. The remaining patient was an infant, asymptomatic at birth, who failed the auditory brainstem response (ABR) test unilaterally at the audiological follow-up at 3 months of age and received antiviral treatment for 6 months.

One or more adverse effects occurred in 10 of the 16 patients (62.5%) who received treatment: 6 cases of neutropenia (median neutrophil count 870/mmc, range 530–920/mmc), 3 of anemia (median Hb value 8.3 g/dL, range 7–8.6 g/dL), 3 of hypertransaminasemia (median ALT 85 U/L, range 58–381 U/L) and 1 of skin rash. In three cases, adverse effects led to early therapy discontinuation, after a median period of 12 weeks: one patient had a recurrent face skin rash and stopped therapy after 9 weeks; the remaining two developed neutropenia after 11 and 15 months respectively. With a median follow-up of 2.2 years (range 0.5–6 years), among the 26 cases asymptomatic at birth only 1 child (3.8%) developed long-term sequelae (SNHL at 3 months of age); no sequelae were reported in the 3 cases of mild disease at birth.

Considering the 12 patients who received treatment within the first month of life, the median follow-up was of 1.5 years: 9 children had a favorable outcome, with no long-term sequelae, whereas no improvement was reported in the 3 cases with SNHL, even though no further hearing deterioration nor other sequelae occurred.

Of the four children treated after the first month of life, three did not improve with therapy, and the neurological and hearing impairment persisted unchanged after a median follow-up of 3 years; for the remaining patient, the treatment proved effective, and the patient, who had developed SNHL at 3 months of age, regained a normal hearing function and presented no long-term sequelae.

Data concerning prior awareness of the infection were available only for 49 of the 59 families referred to our center: 19 families knew about cCMV before pregnancy and were informed about the preventive measures to reduce the risk of congenital infection. Nevertheless, in three cases, awareness followed a previous pregnancy; we therefore calculated an overall awareness before pregnancy of 32.6% (16 families).

## 4. Discussion

Currently, the universal maternal or neonatal screening for CMV is not routinely recommended. Serological maternal screening does not identify the mothers who will transmit the virus; moreover, non-primary maternal infections can easily go undetected. According to a meta-analysis of Kenneson et al. [1], the transmission rate of infection in mothers with prior immunity is around 1.4%, much lower compared to that of 32% in mothers with primary infection. Nonetheless, considering the cCMV birth prevalence increasing with maternal seroprevalence and given the possibility of severe long-term sequelae also in non-primary infections [18,19], missing the diagnosis of cCMV infection in immune mothers cannot be considered negligible [19,20,21,22]. Indeed, in our population, four patients whose mothers had positive immunoglobulin G in the first trimester of pregnancy and who were not tested at birth for cCMV, received a late diagnosis.

Therefore, it is evident that maternal serological screening alone is not sufficient. Universal neonatal cCMV screening, through urine, saliva or DBS specimens, is a much-discussed issue, due to the debatable cost-effectiveness [9,23,24,25]. Naessens et al. [26] proposed the maternal serological screening followed by targeted testing at birth of those neonates whose mothers tested seropositive during pregnancy, stating that this approach could detect up to 82% of all cCMV infections. In our population, considering all the 59 clinical records that we evaluated, this has indeed proven to be an effective strategy, leading to the detection of cCMV infection in 75% of the newborns whose mothers tested positive at the serological screening during pregnancy and who were consequently screened at birth; among these patients, there were five cases of non-primary infection that otherwise would have probably gone undetected.

The signs and symptoms detected in our symptomatic patients group were consistent with those reported in other populations studied, reflecting their usual variety and non-specificity [10,12]. The prevalence of symptomatic cCMV tends to vary in the literature, because this condition lacks a common definition: in our cohort, the percentages of patients symptomatic at birth were significantly different if considering the result of clinical examination alone (17%) or together with radiological and laboratory investigations (36.6%).

Nevertheless, the three cases of congenital cardiopathy are a peculiarity of our population. Only four isolated case reports of cardiomyopathy linked to cCMV infection were reported in the literature [27,28,29,30]. Although CMV is a relatively frequent cause of myocarditis in childhood, studies on the relationship between cCMV and cardiac involvement are lacking.

Considering the early antiviral therapy, started within the first month of life, the management of our patients was in line with the indications of the most recent review on the subject and with the European Expert Consensus Statement on the Diagnosis and Management of Congenital Cytomegalovirus [8,9]. In our small population, we observed similar outcomes to the ones described in the literature with the aim to prove the protective effect of an early therapy against the development of other long-term sequelae or the progression of damage, especially hearing loss [13,15,17]. Even though our population was too small to prove the above-mentioned protective effect, some attention should be paid to the fact that the four cases that were diagnosed late and were not properly treated showed signs of severe CNS involvement on arrival at our center. According to literature data, early diagnosis and therapy might have prevented some of these sequelae. Nevertheless, even an early therapy proved ineffective, in our population, in reverting the damage in patients with SNHL already established at birth.

However, there are still some aspects of treatment that need to be further investigated. The asymptomatic patients and those with mild disease were not treated, according to the current indications, and no long-term sequelae were detected during their follow-up except for one case, who developed SNHL. Our findings are consistent with a recent retrospective analysis by Turriziani Colonna et al. in which, however, treatment with Valganciclovir was performed in both symptomatic and asymptomatic patients [15]. Nevertheless, the current Consensus Statement does not clearly address treatment in these patients. Trials are currently underway in order to identify both benefits, namely, audiological outcomes, and risks of an antiviral treatment in these groups [31]. Evidence is still lacking also about the efficacy of the therapy when started beyond the neonatal period. No RCT are available at the moment, and the only multicenter clinical trial on the treatment with valganciclovir for cCMV-infected infants with isolated hearing impairment has been discontinued due to the lack of enrollment [32]. There are, however, case series on treatment after one month of age, which have reported good outcomes [33,34]. Del Rosal et al. [34] included in their retrospective study infants with cCMV who had started an antiviral treatment with Valganciclovir beyond the neonatal period and, after an average period of six months, the treatment appeared to prevent further deterioration and produce improvement, which was sustained at the 12-month check-up. In our population, a similar favorable outcome was encountered in one of the four cases treated after the neonatal period, in whom hearing impairment developed at 3 months of age, with subsequent improvement and normalization of the hearing level after the antiviral treatment.

Regarding the awareness about the infection, in our population we found that 32.6% of families had heard of congenital cCMV before pregnancy. In a survey carried out by Cannon et al. [5] on a U.S. population, the percentage of awareness was significantly smaller, since only 13% of the interviewed women stated a knowledge of cCMV. A similar questionnaire was proposed by Binda et al. [6] to a large, fairly young and well-educated Italian population, reporting that about 31% of the 10,190 respondents knew about cCMV. Therefore, we can state that the awareness that we encountered was comparable to that of the Italian population and that it was significantly higher than the one found in the study carried out by Cannon et al. [5] among the U.S. population, even though still not satisfactory.

Greater awareness of cCMV infection and its risks also means having a better knowledge regarding hygiene and behavioral measures to prevent said infection, which have proven to be very effective [20]. Therefore, improvements in this area should be pursued.

Some limitations of our analysis should be reported. First, the sample size was limited and comprised patients afferent to a single center. Second, due to the follow-up still being in progress for most of the patients, we cannot confirm the long-term antiviral therapy efficacy nor exclude the occurrence of sequelae in the long term.

In conclusion, the global disease burden of cCMV infection, along with its sequelae, is still considerable. Efforts, therefore, should be made to improve awareness about this congenital infection and the preventive strategies that can be implemented. Information and prevention are indeed our first and most important weapons against infectious diseases. It is also crucial to reach an agreement on the opportunity for a universal cCMV infection screening and its most effective strategy, taking into consideration the evidence already available regarding its potential benefits. Nevertheless, screening should be accompanied by adequate follow-up and therapy strategies. There are indeed a lot of aspects regarding the management of these patients, especially treatment, that are still left to the initiative of single experts, whereas more clinical trials are needed to better determine the timing and candidates for therapy.

Our descriptive study aims at highlighting these difficulties and emphasizing the importance of a much-needed unified approach.

## Figures and Tables

**Table 1 children-09-00655-t001:** Clinical manifestations in 15 children with symptomatic cCMV infection at birth.

Sign or Symptoms of cCMV at Birth	No. (% of Symptomatic Patients)
Abnormal physical examination	8 (53.3%)
*Small for gestational age (SGA)*	7 (46.7%)
*Microcephaly*	2 (13.3%)
*Hepatosplenomegaly*	1 (6.7%)
*Abnormal neurological examination*	1 (6.7%)
Abnormal blood tests	2 (13.3%)
*Anemia*	1 (6.7%)
*Thrombocytopenia*	1 (6.7%)
*Elevated liver enzymes*	1 (6.7%)
SNHL *Monolateral* *Bilateral*	3 (20%) 2 1
Brain imaging anomalies *Cystic lesions* *White matter abnormalities* *Calcifications*	9 (60%) 7 9 1

## Data Availability

The data presented in this study are available on request from the corresponding author. The data are not publicly available due to their containing information that could compromise the privacy of research participants.

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
