# Peer review of "Congenital CMV, Lights and Shadows on Its Management: The Experience of a Reference Center in Northern Italy"

_children, 2022, doi:10.3390/children9050655_

Round 1

Reviewer 1 Report

This is a good and relevant retrospective review of congenital CMV. It provides good data and arguments towards universal screening for cCMV, which has been a topic of debate. It would be interesting to add data on those that were asymptomatic to know whether they then developed late sequelae. Would like clarification on the length of follow-up of the patients diagnosed with cCMV, whether symptomatic or asymptomatic.

Reviewer 2 Report

This manuscript entitled "Congenital CMV, lights and shadows on management: the experience of a reference center in northern Italy” by Dr Rubinacci et al is a retrospective, single center study looking to review its own experience and complement it with a review of literature

Overall, the study is not original as there have been several studies looking at the clinical presentation and outcome of congenital CMV; however what is important is the extreme need to spread the knowledge about the existence of congenital CMV as the author themselves state and the need to unify management (definition and treatment indications) which is different among countries and guidelines are not sustained in many occasions by real evidence-base medicine but more by expert opinions.

I do have some comments though:

Even though it is well written, in some parts the English style could be improved. I would recommend a review from a native English speaking person.

Methods:

-I think it would have been more appropriate to present data as a Median rather then Mean as data can be more accurate

Results:

-Page 3, line 99: What prompted testing these 41 newborns at birth? Would specify with more details, if data is available, the reason for testing: if all because of known seroconversion of mothers or because they were symptomatic or in the asymptomatic newborns just as a random test

-Page 3, Line 127: the 4 late diagnosis were treated with valganciclovir after the first month of life. Apart from 1 case where it is specified that diagnosis was at around 3 months, what was the age at initiation of treatment for the other 3? As it seems that the other 3 were older infants or toddlers. What was the rational to start treatment at this late age (absolutely no data available indicating a benefit) and for how long were these kids treated? 6 weeks or 6 months?

-Page 4, line 135: In the 3 cases where valganciclovir was discontinued due to side effects, which were the exact extreme for discontinuation? Neutropenia? Anemia?

-Page 4, line 142: it is ambitious to state that therapy “proved effective in preventing further hearing deterioration and other sequalae” and I would eliminate that statement. This is a retrospective study and these are only 3 kids and we cannot assume that they would have progressed for sure. This is the main issue we are still facing with cCMV management: there is a profound lack of knowledge about what the right management is

Discussion:

-Page 5, lines 192 to 197: again I would soften the statements about the effectiveness in preventing long term sequelae as the study sample is small, retrospective and not randomized

-Page 5, line 206: The ValEar study (reference 31) was unfortunately discontinued due to lack of enrollment, therefore is not “ongoing” anymore
